# Optimization of 3D Printing Parameters of High Viscosity PEEK/30GF Composites

**DOI:** 10.3390/polym16182601

**Published:** 2024-09-14

**Authors:** Dmitry Yu. Stepanov, Yuri V. Dontsov, Sergey V. Panin, Dmitry G. Buslovich, Vladislav O. Alexenko, Svetlana A. Bochkareva, Andrey V. Batranin, Pavel V. Kosmachev

**Affiliations:** 1Microelectronics of Multispectral Quantum Introscopy Laboratory of the R&D Center “Advanced Electronic Technologies”, National Research Tomsk State University, 634050 Tomsk, Russia; sdu@ispms.ru (D.Y.S.); kosmachev@mail.tsu.ru (P.V.K.); 2Department of Materials Science, Engineering School of Advanced Manufacturing Technologies, National Research Tomsk Polytechnic University, 634050 Tomsk, Russia; doncov@mail2000.ru; 3Laboratory of Mechanics of Polymer Composite Materials, Institute of Strength Physics and Materials Science of Siberian Branch of Russian Academy of Sciences, 634055 Tomsk, Russia; aleksenkovo@ispms.ru (V.O.A.); svetlanab7@yandex.ru (S.A.B.); 4Laboratory of Nanobioengineering, Institute of Strength Physics and Materials Science of Siberian Branch of Russian Academy of Sciences, 634055 Tomsk, Russia; buslovich@ispms.ru; 5Engineering School of Non Destructive Testing, National Research Tomsk Polytechnic University, 634050 Tomsk, Russia; batranin@gmail.com

**Keywords:** ANN, PEEK, glass fiber, polymer matrix composite (PMC), FDM, optimization, Taguchi method, additive manufacturing (AM), computed micro-tomography (micro-CT)

## Abstract

The aim of this study was to optimize a set of technological parameters (travel speed, extruder temperature, and extrusion rate) for 3D printing with a PEEK-based composite reinforced with 30 wt.% glass fibers (GFs). For this purpose, both Taguchi and finite element methods (FEM) were utilized. The artificial neural networks (ANNs) were implemented for computer simulation of full-scale experiments. Computed tomography of the additively manufactured (AM) samples showed that the optimal 3D printing parameters were the extruder temperature of 460 °C, the travel speed of 20 mm/min, and the extrusion rate of 4 rpm (the microextruder screw rotation speed). These values correlated well with those obtained by computer simulation using the ANNs. In such cases, the homogeneous micro- and macro-structures were formed with minimal sample distortions and porosity levels within 10 vol.% of both structures. The most likely reason for porosity was the expansion of the molten polymer when it had been squeezed out from the microextruder nozzle. It was concluded that the mechanical properties of such samples can be improved both by changing the 3D printing strategy to ensure the preferential orientation of GFs along the building direction and by reducing porosity via post-printing treatment or ultrasonic compaction.

## 1. Introduction

Polyetheretherketone (PEEK) is a high-performance high-temperature semi-crystalline polymer with improved physical and mechanical properties, chemical resistance, as well as bioinertness [1]. Respectively, a significant number of researchers have been investigating the features of PEEK [2,3]. However, its high cost greatly narrows the range of applications, primarily towards high-tech industries and medicine.

In the manufacturing of machine parts and friction units, PEEK is typically loaded with fillers [4,5], among which the most widely used are chopped carbon fibers (CCFs) and glass fibers (GFs) at contents of about 30 wt.% [6,7]. Antifriction composites, fabricated by injection molding, contain smaller amounts of reinforcing fibers but they are additionally filled with polytetrafluoroethylene (PTFE) and/or graphite, for instance, for improving their functional characteristics. For such cases, the term ‘high-pressure velocity’ (HPV) composites has been proposed [8].

Recently, 3D printing methods have become widespread for manufacturing numerous products from PEEK-based composites [9], in particular with the use of fused deposition modeling (FDM)/fused filament fabrication (FFF). Typically, neat PEEK is used as a filament matrix with the content of other fillers (like CFs, GFs, AFs, PTFE, graphite, etc.) less than ≤10 wt.%, since melt flow index (MFIs) is sharply reduced at greater concentrations of CCFs or GFs, preventing the formation of dense and uniform structures of additively manufactured products. An alternative approach involves the use of both granular feedstocks and 3D printing heads with microextruders [10], but few research results have been published so far on 3D printing with such high-viscosity thermoplastics [11,12,13,14].

For 3D printing, including with thermoplastic composites, it is necessary to simultaneously optimize a set of technological (input/control) parameters that ensure the high quality of additively manufactured (AM) products. The latter means the maximum precision of shapes, the minimum number of discontinuities, the uniformity of both macro- and microstructure, etc. Their quality is controlled by output (mechanical and structural) parameters. Thus, the 3D printing process can be represented as a black box, and a given set of some functional (output) parameters is achieved by varying the input ones. According to the formulation, such a problem can be solved via the design of experiments [15]. As compared to the other methods: (i) Box Behnken design [16]; (ii) Rotatable Central composite design of experiments (Box-Wilson Central Composite Designs) [17]; (iii) D-optimal design [18]; (iv) Doehlert design [19]; (v) Response surface method (RSM) [20] the Taguchi method [21] is widely applied due to a possibility of multiple reductions in the number of required results in contrast to full-scale experiments [22].

Another effective approach for solving such problems is the use of artificial neural networks (ANNs) [23]. However, their learning requires large training samples, limiting the possibilities of computer simulation of the 3D printing processes [24]. Therefore, it is important to implement methods that enable increasing the sample sizes artificially (non-experimentally) [25]. Enlarging the sample size based on experimental data was carried out by preliminary analysis of the 3D printing process and synthesis of additional preliminary knowledge for the limit values of the process parameters.

Despite numerous reported results and some achieved successes in this field of science, the challenge of 3D printing with high-viscosity thermoplastic composites remains relevant. The reasons are both specifics of AM and complex dependences of the structure formation on a set of input (technological) parameters. This issue requires in-depth research in both materials science and machine learning when ANNs are implemented to solve the problem of approximating a vector quantity in the space of multiple parameters.

The aim of this study was to optimize a set of input parameters (the V_3D_ travel speed, the T_ext_ extruder temperature, and the ν extrusion rate by varying rotation speeds of the microextruder screw) for 3D printing with the high-viscosity PEEK/30GF composite from a commercially available granulate. For this purpose, ANNs were implemented for computer simulation of full-scale experiments.

As a null hypothesis for AM of high-quality products from the PEEK/30GF composite, it was assumed that it is necessary to ensure the maximum material feeding into the printing zone by reducing the V_3D_ travel speed of the moving extruder head by increasing both the T_ext_ extruder temperature and the ν extrusion rate.

The selection of glass fibers as filler material was motivated by their: (i) availability as a commercial grade feedstock; (ii) lower price; (iii) ease of processing; (iv) large interphase adhesion, and (v) possibility of substantially improving mechanical properties of PEEK-based composites. Their input in increasing mechanical properties of PEEK-based composites is a little bit lower in contrast with carbon fibers; however, in terms of affecting 3D printability, they are quite comparable.

The article is structured as follows. Section 2 describes materials and research methods, including the design of the experiment using the Taguchi method. Section 3 presents the results of assessing the mechanical properties and analyzing the structures of the composite, including scanning electron microscopy (SEM) and IR spectroscopy. Section 4 is devoted to the optimization of the 3D printing parameters via computer simulation using ANNs. In Section 5, the obtained results are discussed and some prospects are justified for this scientific direction.

## 2. Materials and Methods

Initially, rectangular shape samples (templates) were fabricated by the FDM method using the ‘PEEK KETASPIRE KT-880 GF30 BG20’ granulate with a length of 3 mm and a diameter of 2 mm (SOLVAY, Shanghai, China) that included 30 wt.% GFs. In this way, an ‘ArmPrint-2’ laboratory 3D printer was deployed (Tomsk Polytechnic University, Tomsk, Russia). It was controlled by the ‘LINUX CNC’ OS operating system and equipped with a single-screw microextruder with a nozzle diameter of 0.5 mm. The ν extrusion rates (amounts of the fed material) were changed by varying rotation speeds of the microextruder screw. The temperature of the heated bed of the 3D printer was 180 C. The 3D printer did not include any thermal chamber, so the air convection of the manufactured samples and their cooling conditions were not instrumentally controlled. However, the temperature conditions were identical in all cases.

Digital model files were created using the ‘Repetir-Host V2.1.3’ software package (Hot-World GmbH & Co. KG Knickelsdorf 4247877, Willich, Germany) and the ‘Slic3r’ slicer (licensed under the GNU Aero General Public License, version 3). The software parameters were preset as the following:The layer height was 0.2 mm;The first layer height was 0.3 mm;The perimeter was 2 lines;The top solid layer was 0 and the bottom one was 0;The infill was 80%;The rectilinear and speed were the same, but the first layer was 50 %; the extrusion width infill was 100%.

For uniaxial static tensile testing, the type 5 dog-bone specimens were cut out according to ISO 527-2:2012 [26]. Their surfaces were processed with sandpapers (grit 240) and conditioned at room temperature for more than 24 h. The tests were carried out using a ‘Gotech Al-7000M’ electromechanical machine at a cross-head speed of 1 mm/min.

The structures of the samples were analyzed with a ‘LEO EVO 50’ SEM (Carl Zeiss, Oberkochen, Germany) at an accelerating voltage of 20 kV. Initially, the notched samples were cooled in liquid nitrogen at −197 °C for one hour and mechanically fractured then. In a vacuum, copper films ~10 nm thick were deposited on the fracture surfaces using a ‘JEOL JEE-420’ vacuum evaporator (JEOL USA, Inc., Peabody, MA, USA). The requirements for the thickness of the conductive films were justified by the need to preserve the morphology of the original fracture surfaces. Then, the filler distributions and the supermolecular structures of the samples were investigated.

The chemical structures of the samples were analyzed by IR spectrometry. IR spectra were recorded using both ‘NICOLET 5700’ (Thermo Fisher Scientific, Waltham, MA, USA) and ‘FT-801’ (SIMEX, Novosibirsk, Russia) Fourier-transform IR spectrometers in the diffuse reflectance range of 600–4000 cm^−1^ with a diamond (Single Reflection Diamond ATR).

Computed microtomography (micro-CT) of the samples was carried out with an ‘OREL-MT’ tomograph assembled at Tomsk Polytechnic University [27]. The tomograph was equipped with a ‘XWT 160-TC’ high-focus X-ray machine (X-RAY WorX, Garbsen, Germany), a ‘PaxScan-2520V’ flat-panel X-ray detector (Varian, Palo Alto, CA, USA), and a research object positioning system. Conical X-ray beam geometry and projection magnification were implemented to increase the spatial resolution of images. The ‘Bruker-microCT’ software package v.1.18 was used for tomographic reconstruction and visualization of the obtained data [28]. The key characteristics of the tomograph are presented in Table 1.

The following micro-CT mode was utilized: an anode voltage of 100 kV, an anode current of 35 μA, an angular step of scanning projections of 0.2 degrees, a number of projections of 1800, and a voxel size of 6.35 μm.

The Taguchi method was used for trial optimizing the 3D printing parameters. The input factors were the V_3D_ travel speed, the T_ext_ extruder temperature, and the ν extrusion rate. Initially, the levels of these input parameters were empirically determined for the sustainable 3D printing process within their wide ranges.

The T_ext_ extruder temperature was preset at 440 °C since this level enabled the melting of the high-viscosity PEEK/30GF composite. It was an important parameter for 3D printing at the V_3D_ travel speeds of about 40 mm/s (to ensure an acceptable production rate). Lower T_ext_ extruder temperatures significantly reduced the 3D printing rate because they did not allow a sufficient amount of the molten composite material to be extruded. So, the T_ext_ extruder temperatures were 420, 440, and 460 °C (Table 2). The V_3D_ travel speeds were preset at 20, 30 and 40 mm/s. As mentioned above, the ν extrusion rates were controlled by varying rotation speeds of the microextruder screw. Initially, a multiplier of 4.5 rpm was determined corresponding to the number of screw revolutions per unit track length upon 3D printing. According to the results of some trial experiments, the shapes of the samples additively manufactured from the PEEK/30GF composite correlated well with their digital models. So, the ν extrusion rates were preset at 4.0, 4.5, and 5.0 rpm (Table 2).

## 3. Results

### 3.1. Mechanical Properties

Table 3 illustrates representative values of the key mechanical properties obtained by averaging over at least four experimental points according to the Taguchi L9 design. A preliminary analysis of these data enabled the conclusion that the ranges of changes in the input parameters used when planning the experiment did not lead to multiple variations in the mechanical properties of the AM samples. In addition, the obtained values were noticeably inferior to the data reported by the feedstock manufacturer since they involved the use of injection molding or compression sintering as a fabrication method [29].

Then, graphs were drawn according to the Taguchi method [10], characterizing the influence of each of the (input) parameters on the mechanical properties of the AM samples. This made it possible to quantify their effect by analyzing the implemented modes (the combinations of the 3D printing parameters) and propose the rational one. For this purpose, the ‘bigger is better’ principle and the signal-to-noise (S/N) ratio were applied. According to this methodology, the maximum S/N ratios characterized the maximum variability of the output parameters (the mechanical properties) and corresponded to the achievement of their maximum values. In this case, such a fact reflected the greater significance of a particular level of a 3D printing parameter.

It should be noted that in contrast to the elastic modulus and the tensile strength used as the output parameters, for which the maximum values had to be achieved, a target value of elongation at break was not obvious according to the ‘bigger is better’ principle. Typically, particulate composites at filling degrees of 25–30% were characterized by brittle failure, corresponding to their high mechanical strength. In this approximation, just a negligible elongation at the break value reflected the preference of the 3D printing mode. On the other hand, the porosity of the PEEK/30GF composite significantly reduced its ductility, being a negative factor. Accordingly, the authors prefer not to discuss in detail the role of the elongation at break values and not to consider it when analyzing the data on the influence of technological factors.

For the ‘ν extrusion rate’ input factor, higher tensile strength and elastic modulus values were achieved when using its third level of 5 rpm (Figure 1a,b, respectively). At the same time, a multidirectional trend was observed for the elongation at break levels (Figure 1c), which confirmed the above reasoning on the inexpediency of taking it into account. Therefore, the maximum ν extrusion rate had to be preset according to the Taguchi method.

An analysis of the influence of the ‘T_ext_ extruder temperature’ input factor on the mechanical properties of the samples showed that their elastic modulus values were characterized by a pronounced rising trend with the maximum at T_ext_ = 460 °C (Figure 1b). For the tensile strength, higher values were also observed at T_ext_ = 440–460 °C (Figure 1a), while the minimum S/N ratio for the ‘elongation at break’ parameter corresponded to the maximum T_ext_ level of 460 °C (Figure 1c). Thereby, the high T_ext_ extruder temperatures had to be preset according to the Taguchi method.

Varying the ‘V_3D_ travel speed’ input parameter from 20 up to 40 mm/s provided higher tensile strength and elastic modulus values at its average level of 30 mm/s, despite they were not significantly lower at the minimum of 20 mm/s. Elongation at break values was characterized by minimal variability at the applied V_3D_ travel speeds. For this reason, the authors again preferred to exclude this “output” parameter from the assessment. Based on the above, the low *V*_3D_ travel speeds had to be preset according to the Taguchi method.

Finally, the Delta (Δ) influence degrees of the input (technological) parameters were ranked according to the Taguchi principle to ensure higher mechanical properties of the samples (Table 4). From the tensile strength perspective, all three factors had comparable Δ influence degrees of 0.804–0.879. For the elastic modulus, the T_ext_ extruder temperature was two times ‘more important’ of a factor than the other two. In the case of elongation at break, the maximum Δ influence degree of 0.853 was exerted by the V_3D_ travel speed, while it was the minimum (Δ = 0.175) for the T_ext_ extruder temperature.

As a preliminary discussion, it should be noted that the above findings were somewhat correlated with the results of Deng et al. [30], who applied the Taguchi method for optimizing the 3D printing mode for neat PEEK (its MFI was surely lower than that of the PEEK/30GF composite). In that case, the highest tensile strength of 40 MPa was achieved at the maximum T_ext_ extruder temperature of 370 °C and V_3D_ travel speed of 60 mm/s, while, the highest elastic modulus of 1563 MPa was obtained at a lower T_ext_ level of 360 °C but the same maximum V_3D_ travel speed of 60 mm/s. The greatest elongation at break value of 14.3% was observed at the minimum V_3D_ travel speed of 20 mm/s. These maximum mechanical properties were noticeably inferior to the cast polymer. In [30], it was stated that 3D printing angles, nozzle diameters, and bead widths had to be optimized as well.

So, the results of the above analysis carried out within the framework of the adopted Taguchi method did not allow both to determine the exact (but not applied in the experiments) values of the input parameters and to give an unambiguous interpretation of the reason for their Δ influence degrees. For clarifying these nuances and assessing the ‘3D printing mode–structure–properties’ relationship, the micro- and chemical structures of the samples were examined by SEM and IR spectroscopy, respectively.

### 3.2. Microstructures

Appendix A shows lower magnification SEM micrographs of the PEEK/30GF composites additively manufactured using the FDM modes presented in Table 2. Note, that no signs of agglomeration of GFs were observed. The pattern of the fracture surfaces (obtained after exposing the samples to liquid nitrogen) was not brittle. Generally, this fact was consistent with the relatively high elongation at break values of ~5%, since similar PEEK-based composites possessed minimal ductility (~1%) at comparable filling degrees. On the other hand, the presented SEM micrographs did not reveal any fundamental differences in the “macrostructures” of all investigated samples.

An analysis of the SEM micrographs”take’ at higher magnification (Figure 2) enabled the conclusion that there was no predominant orientation of GFs along the direction of laying the molten filament, despite the FDM method used for 3D printing, which involved the extrusion of the molten granular feedstock. The reason could be the fact that GFs were stochastically oriented in the initial granulate.

Nevertheless, the more fundamental identified result was high porosity levels, reaching up to dozens vol.%. In the Discussion section, estimates of the effect of porosity on the elastic modulus of the PEEK/30GF composite are given, explaining its low level at the sufficiently high filling degree of reinforcing GFs.

The authors did not attempt to statistically estimate the porosity levels for each of the nine samples. The reason was the fact that the elongation at break values, which were very sensitive to porosity, did not change within noticeable limits. For this reason, one could conclude that it was inappropriate to determine a correlation between the porosity and the set of the 3D printing parameters.

In addition to the porosity statement and the approximate estimates of its levels given above, it was important to note the dimensions of pores in order to discuss the aspect of their formation, including in terms of the applied 3D printing parameters. According to Figure 2, pores were generally isolated (not through), while their characteristic sizes (with a predominantly round shape) were several tens of microns. At the same time, the pores were not predominantly formed near reinforcing GFs.

Thus, the revealed high porosity levels indicated that some thermal destruction of the polymer could occur at the elevated T_ext_ extruder temperatures of 420–460 °C (recall that the melting point of PEEK was ~343 °C). To verify this fact, the results of the IR spectral analysis of the samples additively manufactured at different T_ext_ extruder temperatures are discussed below.

Note that it was difficult to ensure uniform spreading of the high-viscosity molten composite containing fibrous inclusions, which differed significantly in rheological properties, upon 3D printing at high both ν extrusions rates and V_3D_ travel speeds. Loading PEEK with 30 wt.% GFs necessitated a higher extrusion pressure, so the 3D printer equipped with the microextruder was utilized. For this reason, it could not be expected that the formed structures were highly dense and continuous. This fact was analyzed in more detail by computer simulation using ANNs, the results of which are reported below in Section 4.

### 3.3. Chemical Structures

Appendix A presents the IR spectra of the samples of the PEEK/30GF composite additively manufactured at the T_ext_ extruder temperatures of 420, 440, and 460 °C. The key task was to show that short-term heating up to 460 °C, at which the higher mechanical properties of the samples were achieved, did not contribute to the thermal destruction of the polymer matrix. Such a statement was based on the fact that the given IR spectra differed little from each other even for modes 7 and 9, characterized by the T_ext_ extruder temperature of 460 °C.

## 4. Application of ANNs for Simulation of the 3D Printing Process

Since the study did not allow for the explicit determination of the exact optimal values of technological parameters, partly because their preset ranges may not have been wide enough, ANNs were implemented for computer simulation of the 3D printing process. Since the sample size of nine experimental points was ultra-small, several additional experiments were carried out.

In this section, the statement of the research aim could be formulated as follows. As noted above, computer simulation was an effective tool for optimizing the 3D printing parameters, i.e., improving the mechanical properties with a minimum number of full-scale experiments. Previously, a similar problem was solved by the authors for ultrasonic welding of laminated composites [25]. The proposed methodology involved solving two key issues: (i) synthesis of a single nonlinear model of 3D printing as a process with many inputs and outputs, and (ii) the addition of a priori knowledge to ensure acceptable predictive accuracy of computer-aided simulation using ANNs.

### 4.1. Experimental Data Analysis

In addition to the nine modes (combinations of the 3D printing parameters) analyzed above by the Taguchi method, six additional modes No. 10–15 were tested (Table 5; Figure 3) in order to expand the training sample. As noted above, four to seven laboratory experiments were performed for each set of the input parameters (75 in total), as well as the same numbers of the output ones were determined (tensile strength, elastic modulus, and elongation at break). An illustration of the distribution of the (input) parameters in a three-dimensional space is shown in Figure 3.

The obtained data were analyzed by drawing dependencies of the mechanical properties of the samples of the PEEK/30GF composite on the 3D printing parameters (Figure 4a), interdependencies of their mechanical properties (Figure 4b), and calculating regression models for each of them. Among the obtained data, six laboratory experiments were identified and excluded from a subsequent analysis, the results of which were outside the range of confidence intervals (at a significance level of 0.05).

The assumption of a significant nonlinear dependence of the mechanical properties of the samples on the 3D printing parameters was observed in Figure 4a, which was confirmed by both low coefficients of determination values and high standard errors of multiple linear regression models (Table 6).

### 4.2. Analysis of Priori Knowledge

#### 4.2.1. Prerequisites for a Priori Knowledge

A preliminary analysis of both 3D-printing parameters and predicted mechanical properties of the samples was based on previously acquired knowledge about the features of the analyzed process:The 3D printing process could not be implemented under one of the following conditions: the T_ext_ extruder temperature was below the melting point of the polymer matrix (mode 16), the absence of the fed material (at ν = 0 rpm, mode 20 or V_3D_ = 0 mm/s, mode 18). So, the mechanical properties of such samples were assumed to be zero.It was also assumed that the mechanical properties were lower at an excessively high T_ext_ extruder temperature of 560 °C (mode 17) due to possible thermal degradation of PEEK. The same applied to high V_3D_ travel speed of 60 mm/s (mode 19) and ν extrusion rate of 8 rpm (mode 21), based on the assumption that an excessive amount of the extruded polymer did not allow a homogeneous structure to form.

The mechanical properties of the samples of the PEEK/30GF composite, summarized on the basis of a priori knowledge, are presented in the Appendix A section.

#### 4.2.2. Selecting Acceptable Minimum Values

Then, the authors expertly formulated the requirements for the minimum acceptable values of the output parameters (the mechanical properties according to Table 7). Checking the results of the laboratory experiments for compliance with these acceptable values showed that there was not a single mode among the applied ones (Table 5) that would meet all the requirements. However, modes 4, 5, 7, 8, and 10 were identified, for which the acceptable minimum values were achieved in at least one experiment (Figure 5).

### 4.3. ANN-Based Computer Simulation of the 3D Printing Process

In this study, the authors used two types of simple ANNs that provided nonlinear computer simulation: feedforward networks (FFNNs) and radial basis function networks (RBFNNs). Their inputs were the 3D-printing parameters, while the outputs were the mechanical properties. The selection of both architecture and parameters was described in detail previously [25,31,32], so the applied approach corresponded to that reported in [25].

In the first stage, a sample containing the verified experimental data was used, i.e., a total of 15 vectors of the 3D printing parameters (modes) and 66 vectors of the corresponding mechanical properties. For the FFNN model, the number of hidden layer neurons (from 3 to 8) and activation functions (linear, hyperbolic tangent sigmoid, log-sigmoid) were varied. Upon their learning, synthesized property vectors formed from the experimental values with the addition of pseudo-random numbers with zero mathematical expectation and normalized standard deviation equal to 0.01 were added to the training sample. Computer simulation was carried out many times with an enumeration of the main parameters of the ANNs. For the RBFNN model, the ‘spread’ parameter varied from 0.1 to 0.9, while the ‘goal’ was changed within the 0.100–0.001 range. The results of assessing the areas of suboptimal parameters (SOPs) of the 3D printing process (Table 6) are shown in Figure 6a and Figure 7a for the RBFNN and FFNN models, respectively.

In the second stage, the training sample included the experimental data and a priori knowledge. In order to estimate the influence of training sample size, its effect on the accuracy of model construction for the FFNN was carried out at variation of the number of vectors of input parameters as well as the amount of synthesized data, Figure 8. During the synthesis of the models, the mean square error (MSE) was utilized as the optimization function. It allowed for the characterization of the deviation of the model from the training set. The mean absolute deviation (MAD) of the normalized model values over priory values not taken/specified/determined in the training set was utilized for characterizing the prognosis capability of the models. The verification of the FFNN models over the data of priory knowledge planes has shown quite different dependence: the presence of MAD maximum at the sample size of 69 modes (vectors of the input parameters) and 120 vectors of the corresponding mechanical properties. At the same time, for the FFNN model, the training sample was enlarged by synthesized data with characteristics similar to those at the first stage (see above). It might be concluded that the latter has increased the prognosis capabilities of the developed models. The best characteristics of the ANNs were sought in the same ranges, so two models were selected, for which SOP areas were characterized by the simplest structures (Figure 6b and Figure 7b).

Table 8 presents the statistical characteristics of the learning results for the selected models. Comparing them with the data obtained using the regression model (Table 6), it has been concluded that computer simulation using the ANNs, due to its nonlinearity, was characterized by a low mean square error (MSE) but a high determination coefficient (R^2^). The use of a priori knowledge made it possible to increase the determination coefficient, i.e., reduce the unexplained variance of the model. However, the root-mean-square error increased, which was a natural result of enhancing the modeling area in the extrapolation zone.

For the RBFNN model (Figure 6a), the SOP area was finite and closed. The 3D printing modes, which partially satisfied the optimality condition, turned out to be outside its boundaries. The addition of a priori knowledge to the training sample led to a slight enlargement of the SOP area, compliance with the modes that partially satisfied the target mechanical properties, and significant complication of its shape (Figure 6b).

The SOP area, drawn according to the FFNN model after its learning based only on the experimental and synthesized data, was not limited to four planes: high values of both T_ext_ extruder temperature and ν extrusion rate, as well as both low and high levels of the V_3D_ travel speed (Figure 7a). Such behavior was caused by the location of the satisfactory experimental results on the boundaries of the SOP area and the poor ability of the FFNN model to be extrapolated. Respectively, such a model predicted the target mechanical properties at the boundaries of the analyzed data area and could not be considered adequate. Adding a priori knowledge in the field of extrapolation to the training sample made it possible to limit the SOP area throughout the entire volume and localize the predicted optimal parameters (Figure 7b).

Within the SOP areas, the 3D printing parameters were sought, at which the mechanical properties were maximum. The weighted root-mean-square value was chosen as the maximum search measure. The search results are presented in Table 8. Excluding the “FFNN 66 + 132” model from consideration, all other ones enabled the prediction of the optimal values of both T_ext_ extruder temperatures of 450–460 °C and ν extrusion rates of 4.1–4.59 rpm. A noticeable discrepancy in the predicted data was observed only for the V_3D_ travel speeds (from 13.92 up to 30.91 mm/s), which could be explained by the difference in the general simulation approaches.

The RBFNN model was based on the concept of minimizing approximation in each value of the training sample and searching for the minimum predicted values in the extrapolation zones. Therefore, it was obvious that it predicted the optimal values close to the known parameters in all cases. Conversely, the root-mean-square minimization was applied in the FFNN model to identify general trends in the training sample. This algorithm could lead (as was shown in the studied case) to the removal of the predicted optimal parameters from the region of known values.

The predicted 3D printing parameters were not duly verified for several reasons. Firstly, they differed slightly from the results of the laboratory experiments, enhancing the probability of obtaining similar data. Secondly, the results of the applied Taguchi method (Section 3) were in qualitative agreement with the data obtained by computer simulation using the ANNs. Thirdly, the achieved mechanical properties of the additively manufactured samples were lower than the values given by the manufacturer of the PEEK/30GF feedstock for injection molding, so the reasons for this phenomenon are discussed below.

## 5. Discussion

### 5.1. Computer Simulation on the Effect of Porosity on the Mechanical Properties of the PEEK/30GF Composites

Since it was convincingly shown above that the formation of porosity could significantly reduce the mechanical properties of the PEEK/30GF composite, the influence of pores (air inclusions) of various diameters (from 20 up to 100 μm) on the elastic modulus was assessed by computer simulation using the finite element method (FEM). It should be noted that the studied effect of lowering the mechanical properties of polymer composites with increasing porosity was also reported elsewhere [33,34].

The problem of determining the mechanical properties of porous neat polymers and their composites could be reduced to calculating the stress-strain state parameters of a representative volume under uniaxial tension. In this way, the equations of solid mechanics [35,36,37] were solved with subsequent averaging of the obtained values [38].

In this study, the problem of the plane-stress state of the porous PEEK/30GF composite was solved to obtain quantitative estimates of the changes in its mechanical properties. Square computational domains with round pores were applied. Figure 9 shows ones as finite element meshes of 500 × 500 µm in size with a pore content of 30% (the air filling degree) of different diameters (from 20 to 100 µm). The centers of pores were specified using a random number generator, so they were located as uniformly as possible over the computational domains.

In the laboratory experiments, it was possible to visually estimate both characteristic pore sizes and porosity levels with certain errors (Figure 2). Therefore, large or small pores of identical sizes were varied in the calculations (Figure 9). In addition, a combination of them was used being typical for the studied samples. At the same porosity level, the number of small pores was evidently greater than that for large ones.

Lengths of the sides of the representative volume were 1000 µm for pores with a radius of 100 µm, but they were 500 µm for ones with a diameter of 20 µm, considering the convergence of the calculated elastic modulus values within 5%. The mechanical properties of the matrix in the representative volume were taken on the basis of the data reported for the PEEK/30GF composite [39]: the elastic modulus was 7000 MPa, the tensile strength was 80 MPa, and the Poisson’s ratio was 0.39. The boundary conditions corresponded to the stretching of the computational domain along the X-axis, so the displacement along the Y-axis was specified at the top boundary. The bottom boundary was rigidly fixed and the displacements were zeroed along the X and Y axes. The lateral boundaries were free, so both normal and shear stresses were set to zero.

Figure 10 shows slightly nonlinear dependences of equivalent elastic modulus values versus the content of pores of various sizes. With the same porosity, the elastic modulus was lower in the presence of large pores than in the case of small ones. The discrepancy between the curves and their nonlinearity could be caused by the distribution of pores over the computational domain with increasing porosity levels. When pores of different sizes were present, with a fairly uniform distribution over the computational domain, the elastic modulus values were approximately in the middle between the two curves shown in Figure 10.

Figure 11a,b shows stress distributions over the representative volume with the dimensions of 1000 × 1000 µm in the presence of pores with diameters of 20 and 100 µm, respectively, at the porosity level of 30%. In addition, Figure 11c presents similar data for the representative volume 500 × 500 µm in size for pores of different diameters. Inside such pores, stresses were zero, but they were maximum at their boundaries in the matrix. The locations of the distribution of the maximum stresses near the pores corresponded to the solution of the Kirsch problem of stretching a plate with a round hole [40]. Theoretically, the magnitude of the stresses depended on the ratio of the pore radius to the distance between them in the elastic case: the smaller this ratio was, the lower the stress in the matrix at the same strains and, accordingly, the lower the elastic modulus was. Considering the nonlinear behavior of the PEEK/30GF composite, the real values were lower.

Based on the above, the low experimental values of the elastic modulus of the PEEK/30GF composite relative to the data reported by the feedstock manufacturer were reliably explained by the formation of pores in the AM samples.

### 5.2. Computed Micro-Tomography (Micro-CT)

Since the authors noted porosity as the key factor in reducing the mechanical properties of the studied samples, and in light of the use of additional modes 10–15 (Table 5), some of them were examined by the micro-CT. As discussed above, increasing the T_ext_ extruder temperature was supposed to reduce the molten polymer viscosity, so selected modes for the analysis differed primarily in this parameter, in particular mode 12 (T_ext_ = 380 °C/V_3D_ = 40 mm/min/ν = 5 rpm), mode 14 (T_ext_ = 440 °C/V_3D_ = 10 mm/min/ν = 5 rpm), and mode 10 (T_ext_ = 460 °C/V_3D_ = 20 mm/min/ν = 4 rpm).

The micro-CT analysis was carried out near the fracture surfaces of the dog-bone specimens with a height of the tomography area of ~8 mm. Three-dimensional views of the examined areas are shown in Figure 12, both from the side of the print head (Figure 12a–c) and from the side of the supporting table (Figure 12d–f). According to these data, no macro-discontinuities were observed, being characteristic of the FDM method due to the sequential deposition of melted filament beads.

Figure 13 presents visualizations of the orthogonal projections of the PEEK/30GF composite to illustrate their internal (meso)structures. On the one hand, GFs were quite densely distributed throughout the bulk samples without a predominant orientation due to the high filling degree, which was noted above when analyzing the SEM micrographs of the fracture surfaces (Figure 2). At the same time, a denser structure of the sample was characteristic of mode 12 at the minimum T_ext_ extruder temperature (Figure 13d,g). In this sample, delamination was observed on the side of the supporting table (Figure 13a), which significantly weakened the section. According to the authors, this fact was the reason for reducing its mechanical properties. Note that the two other technological factors also affected the structures, so only their complex influence is to be discussed.

A less ‘loose’ (meso)structure with some ‘variable-density’ areas was observed for the intermediate T_ext_ extruder temperature of 440 °C (Figure 13b), while a more uniform and denser one throughout the analyzed volume was characteristic of the sample additively manufactured using mode 10 (Figure 13c). The pattern of the structures according to the other two orthogonal projections (Figure 13d–i) to a lesser extent reflected the difference in the above-discussed mechanical properties of these samples. Nevertheless, the sample’s corners could contain large discontinuities, affecting the mechanical properties as well.

Thus, at the minimum T_ext_ extruder temperature of 380 °C, voids were found on the upper side of the sample (Figure 13d,g). As it increased, the samples possessed the more uniform macrostructures (Figure 13e,f,h,i). This fact indicated that the scatter in the mechanical properties revealed in the study could be caused by the macro-heterogeneity of the sample structures. In addition, it was consistent with the results of measuring the cross-sectional areas depending on the position of the section in height for the three samples (Figure 14). In these cases, the greatest dispersion was characteristic of mode 12, while the most negligible one was for mode 10. These results were contributed by both porosity and sample distortions, i.e., deviation from a given rectangular section.

The above data characterized both separate scales of the structures and ‘randomly’ examined sections. Using the embedded software package, a quantitative analysis of the porosity levels was carried out throughout the entire investigated volumes. For the correctness of the calculation, volumes of interest (VOIs) were selected (cut out) and the porosity levels were assessed inside them (an example is shown in Figure 15).

For example, Figure 16 shows the sections used for assessing the porosity levels by applying the following analysis technique. Firstly, a threshold was selected according to which the tomograms were binarized (as a set of sections). This procedure solved the segmentation problem, where black areas corresponded to pores, and white ones reflected to the solid material. The ratios of the cross-sectional areas or volumes characterized the porosity levels presented in Table 9, decreasing with enhancing the T_ext_ extruder temperature.

The authors were aware that the approach applied for interpreting the obtained data, which was based on the dominant role of porosity, was idealized. However, it enabled the analysis of them and suggested a future research direction to improve the mechanical properties of high-viscosity thermoplastics, namely the porosity reduction. They also believed that it was important to once again discuss the possible influence of the orientation of GFs in the initial feedstock.

Figure 17 shows the micro-CT data of the original PEEK/30GF granulate, which enabled the conclusion that the vast majority of GFs were oriented along its axis. However, no preferential orientation of GFs was observed after 3D printing (Figure 13).

### 5.3. Effects of the 3D Printing Parameters

As a concluding comment, the authors return to the issue of the influence of the 3D printing parameters on the structures and the mechanical properties of the AM PEEK/30GF composites (a high-viscosity one, from the extrusion point of view).

It should be noticed, that the optimization of studied technological parameters did not allow to drastically increase the mechanical properties, including with the use of ANN simulation. However, it does not mean that they did not affect the structure and functional properties. The above-cited literature on FDM of PEEK and PEEK-based composites did focus on variation of the same 3D-printing parameters [30]. However, the use of low-viscosity polymers allowed for the reduction of the porosity and attained a wider range of variation of mechanical properties. Thus, the authors were able to compare the results of this study with those reported in the relevant papers [11,12,30].

Since it was not possible to avoid the high porosity of the 3D printed PEE/30GF composites by varying the input parameters with the use of an available 3D printer, the prospect might be the application of post-build processing, i.e., ultrasonic compaction with the use of US-welding machine. In this regard, the developed approach to ANN simulation will be efficient both to find out predicted local values of optimum parameters and to construct estimates of their range.

#### 5.3.1. Extruder Temperature

The influence of this input parameter was most obvious. With its increasing up to 460 °C, the polymer viscosity decreased. More intense interlayer and intermolecular diffusion had to stimulate the mutual penetration of segments of macromolecules upon 3D printing. Due to the enhanced interlayer adhesion, the mechanical properties of the samples had to be improved. Rising the T_ext_ extruder temperature above a certain threshold level could cause thermal destruction of the polymer, which was incorporated into a priori knowledge for computer simulation using the ANNs.

#### 5.3.2. Extrusion Rate

This factor could exert a dual effect on the structures and the mechanical properties of the additively manufactured samples. On the one hand, the PEEK/30GF composite had to possess a nonlinear viscosity change depending on the ν extrusion rate [41]. As a result, increasing the screw rotation speed could lead to the structuring of the molten polymer flow, while enhancing the internal energy of its segments could be accompanied by conformations of the supermolecular structure elements, improving the mechanical properties of the samples. On the other hand, the extruded molten material could be laid out more uniformly near the nozzle exit at low ν extrusion rates, providing more uniform structures and improving the mechanical properties.

#### 5.3.3. Travel Speed

The V_3D_ travel speed factor to a certain extent was similar to the previous one since it determined the amount of the molten material squeezed out from the microextruder at a time. If it was assumed that denser structures were provided at low V_3D_ travel speeds, then a decrease in this parameter had to improve the mechanical properties of the AM samples. Nevertheless, its increasing reduced cooling rates of the additively manufactured samples, stimulating better spreading of the molten polymer, including when depositing a subsequent layer or a nearby bead. In such cases, enhancing the V_3D_ travel speed had to improve their mechanical properties.

## 6. Conclusions

The effect of the 3D printing parameters on the structures and the mechanical properties of the samples manufactured by the FDM method from the high-viscosity PEEK/30GF composite feedstock was investigated. It was shown that the formed macro- and microstructures limited the achievement of their high levels. In particular, the following conclusions were drawn.

The high viscosity of the molten polymer contributed to the great porosity levels of the AM samples which was proven in detail by the SEM micrographs. The presence of pores reduced the elastic moduli below 3 GPa, while the elongation at break values of ~5% were significantly higher than those for similar compression molded (hot-pressed) composites.By computer simulation using two types of ANNs, the optimal combinations of the 3D printing parameters were determined for the PEEK/30GF composite: Extruder temperature: 450 ÷ 460 °C, Extrusion rate: 4.1 ÷ 4.59 rev/min, Travel speed; 13.92 ÷ 30.91 mm/s. When an ultra-small sample was utilized, the application of priory knowledge made it possible to achieve the appropriate simulation accuracy (MSE~0.06). In doing so, the RBFNN was able to construct a more realistic model, while for the FFNN the reasonable limitation of its size was implemented.An interpretation of the differences in the results, predicted using the RBFNN and FFNN models, was proposed based on their operation principles. Generally, the determined optimal values were consistent with those obtained by the Taguchi method, and physically corresponded to the assumption formulated as the null hypothesis: higher quality of AM products from the PEEK/30GF composite was ensured by maximizing material feeding into the 3D-printing zone by reducing the V_3D_ travel speed of the moving extruder head with increasing both the T_ext_ extruder temperature and the ν extrusion rate.The effect of porosity on the mechanical properties of the additively manufactured PEEK/30GF composites was assessed by implementing the FEM-based models of small, large, and mixed pores. The obtained results made it possible to explain the experimentally revealed relatively low level of strength.The applied 3D printing parameters, primarily the T_ext_ extruder temperature, did not lead to a change in the chemical structure (in terms of oxidation) of the polymer matrix and can be used for additive manufacturing of products from the PEEK/30GF composite by the FDM method.The micro-CT analysis of the AM samples enabled the conclusion that the optimal 3D printing parameters were the T_ext_ extruder temperature of 460 °C, the V_3D_ travel speed of 20 mm/min, and the ν extrusion rate of 4 rpm. These values correlated well with those obtained by computer simulation using the ANNs. In such cases, the homogeneous micro- and macro-structures were formed with minimal sample distortions and porosity levels within 10 vol.%.The most likely reason for the great porosity levels was the expansion of the molten polymer when it was squeezed out from the microextruder nozzle of the 3D printer since the pressure in the chamber was caused by its high viscosity. Probably, the mechanical properties of such samples can be improved both by changing the 3D printing strategy to ensure the preferential orientation of GFs in the building direction and by reducing porosity via post-build treatment or ultrasonic compaction. The following research by the authors will be devoted to the implementation of these methods.

## Figures and Tables

**Figure 1 polymers-16-02601-f001:**
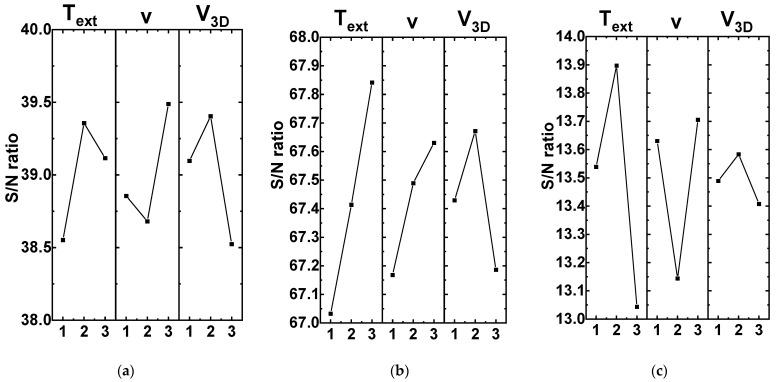
The S/N ratios for different levels of the technological parameters: (**a**) tensile strength; (**b**) elastic modulus; (**c**) elongation at break.

**Figure 2 polymers-16-02601-f002:**
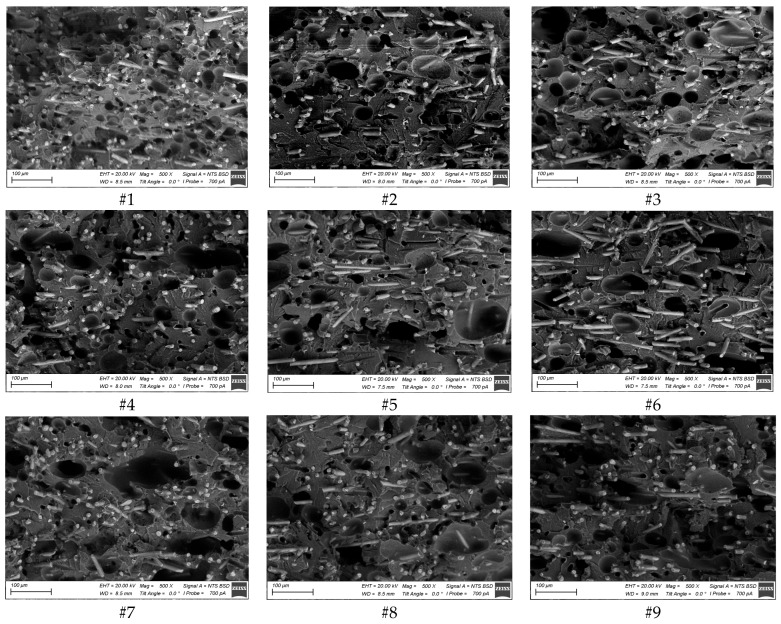
The SEM micrographs of the PEEK/30GF composites additively manufactured using the modes presented in Table 2.

**Figure 3 polymers-16-02601-f003:**
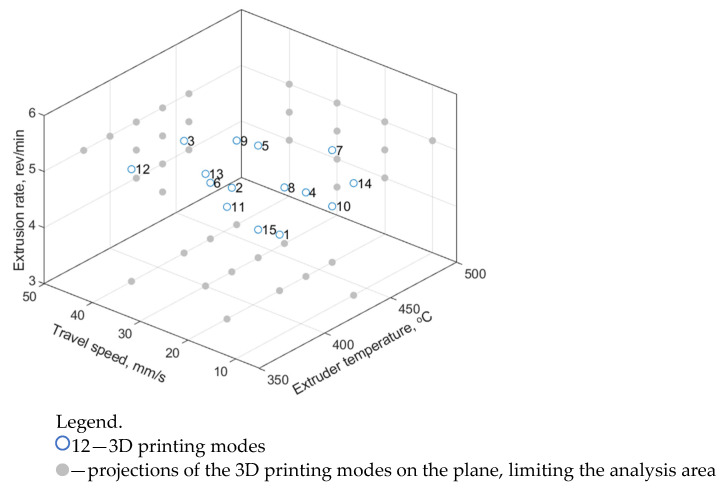
The 3D printing modes of the laboratory experiments in the space of the (input) parameters.

**Figure 4 polymers-16-02601-f004:**
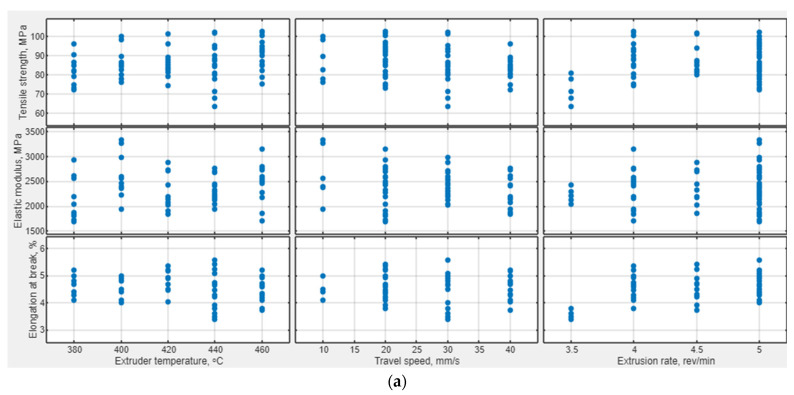
The dependences of the mechanical properties of the samples of the PEEK/30 GF composite on the 3D printing parameters (**a**), as well as both dependences and histograms (**b**) after verification.

**Figure 5 polymers-16-02601-f005:**
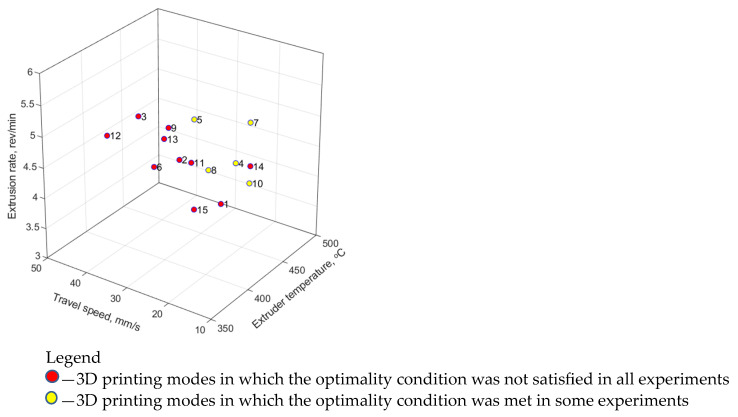
The parameters’ space and the result of checking the 3D-printing modes for compliance with the minimum acceptable property values.

**Figure 6 polymers-16-02601-f006:**
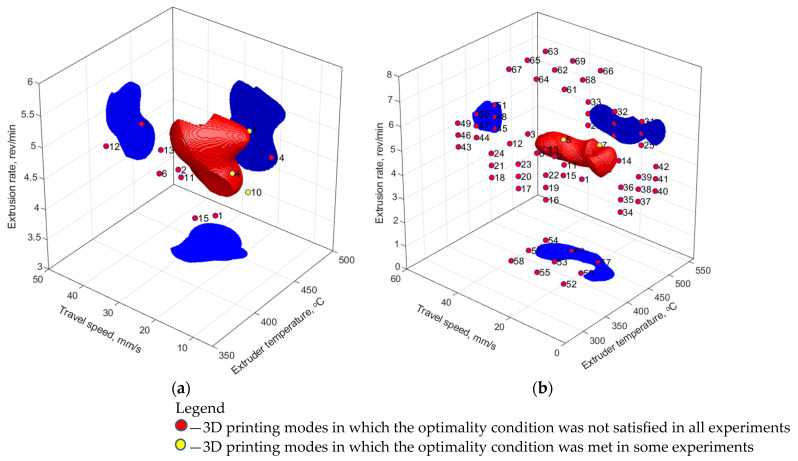
The 3D printing modes and a priori knowledge, as well as the SOP area, drawn using the RBFNN model: (**a**) spread = 0.3, goal = 0.001, the training sample size of 66 vectors; (**b**) spread = 0.3, goal = 0.01, the training sample size of 66 experimental + 54 a priori vectors.

**Figure 7 polymers-16-02601-f007:**
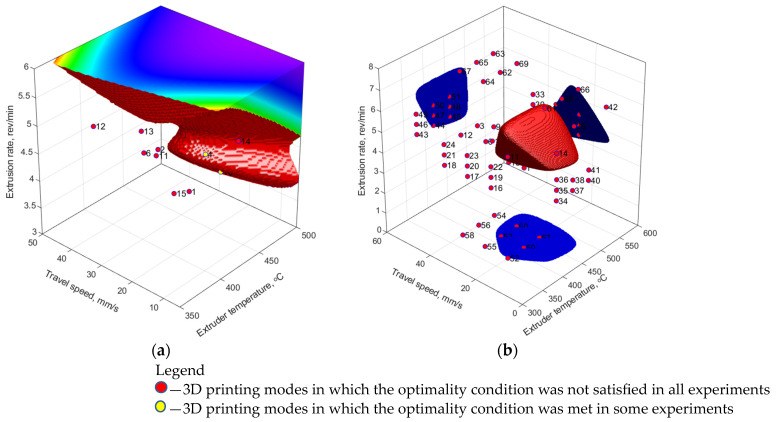
The experimental modes and a priori knowledge, as well as the SOP area, drawn using the FFNN model: (**a**) 4 hidden layer neurons, the sample size of 66 experimental + 132 synthesized vectors; (**b**) 6 hidden layer neurons, the sample size of 66 experimental + 54 prior + 240 synthesized vectors.

**Figure 8 polymers-16-02601-f008:**
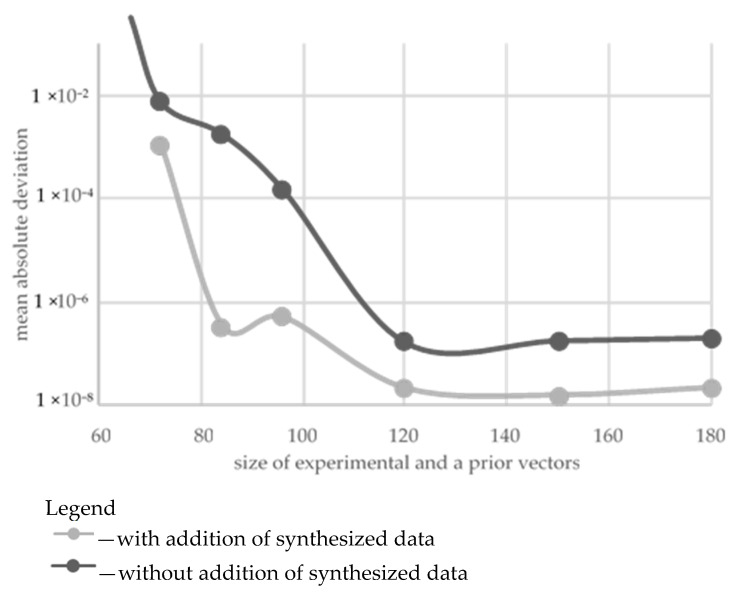
Results of models’ verification within priory knowledge planes as a function of the size of experimental and prior vectors of the properties.

**Figure 9 polymers-16-02601-f009:**
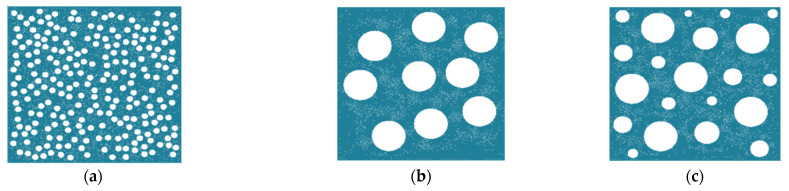
Schematic locations of pores with the diameters of 20 µm (**a**), 100 µm (**b**), and from 20 to 100 µm (**c**) in the computational domains at the porosity of 30%.

**Figure 10 polymers-16-02601-f010:**
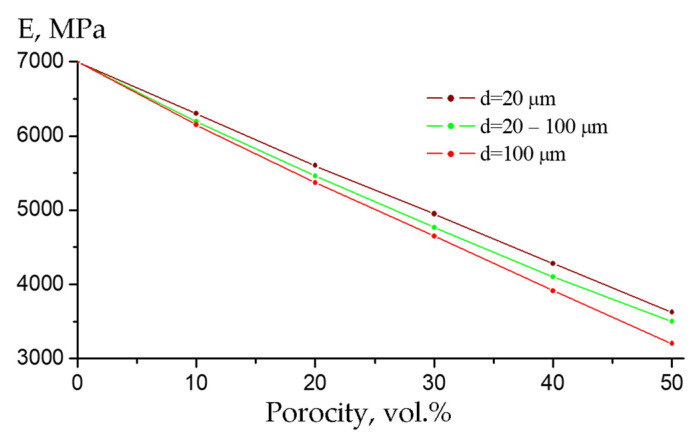
The elastic modulus versus porosity dependences for pores with different diameters *d*.

**Figure 11 polymers-16-02601-f011:**
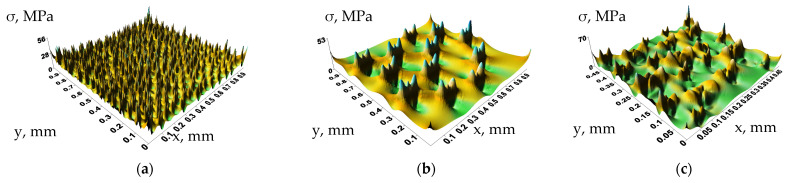
The stress distribution surfaces over the representative volume in the presence of pores with the diameters of 20 µm (**a**), 100 µm (**b**), and from 20 to 100 µm (**c**) at a porosity of 30%.

**Figure 12 polymers-16-02601-f012:**
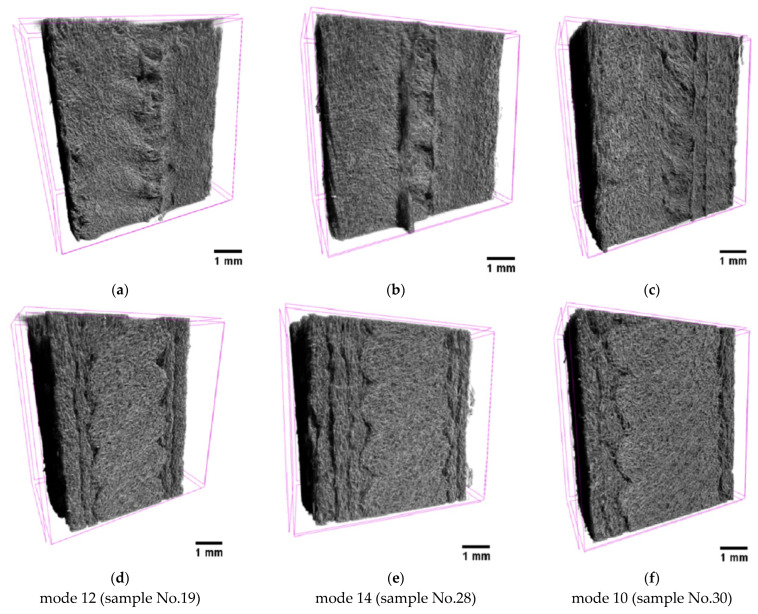
The three-dimensional micro-CT views of the samples, from both supporting table (**a**–**c**) and 3D-printing head (**d**–**f**) sides; mode 12 (**a**,**d**); mode 14 (**b**,**e**); mode 10 (**c**,**f**).

**Figure 13 polymers-16-02601-f013:**
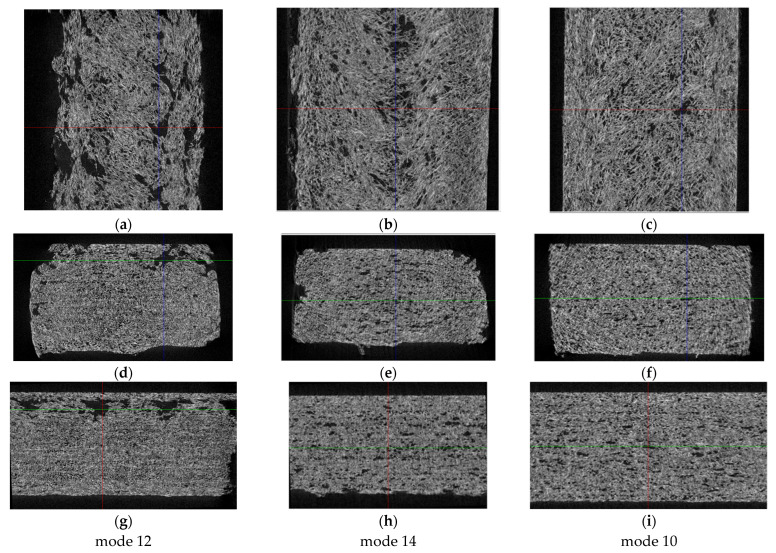
The orthogonal projections of the samples near the fracture surfaces at the image (slice) sizes of 7.5–8.0 mm (**a**–**c**), 7.5–4.5 mm (**d**–**f**), and 8.0–4.5 mm (**g**–**i**). Red denotes Z- axis section; Blue denotes X-axis section; Green denotes Y-axis section.

**Figure 14 polymers-16-02601-f014:**
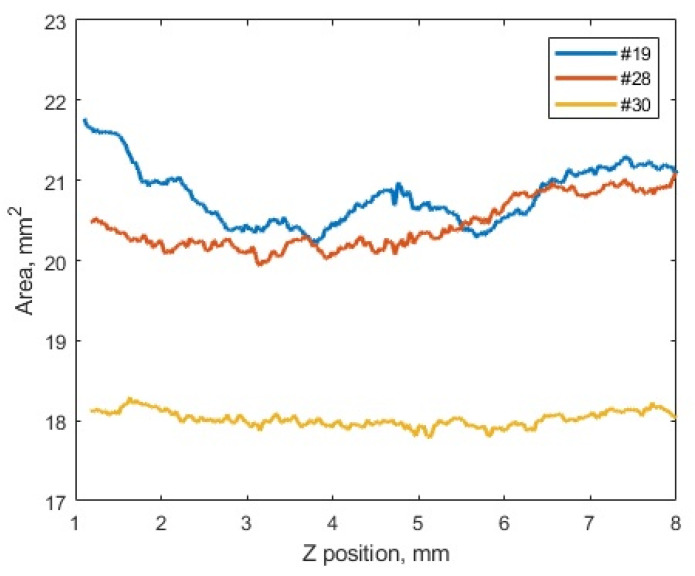
The comparative results of assessing the cross-sectional areas of the samples depending on the position of the height section (along the Z axis): mode 12 (sample No.19); mode 14 (sample No.28); mode 10 (sample No.30).

**Figure 15 polymers-16-02601-f015:**
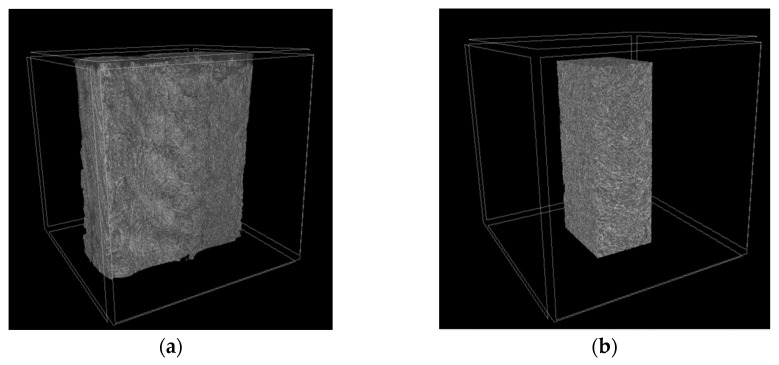
Visualizations of a full tomogram (**a**) and a cut-out VOI (**b**).

**Figure 16 polymers-16-02601-f016:**
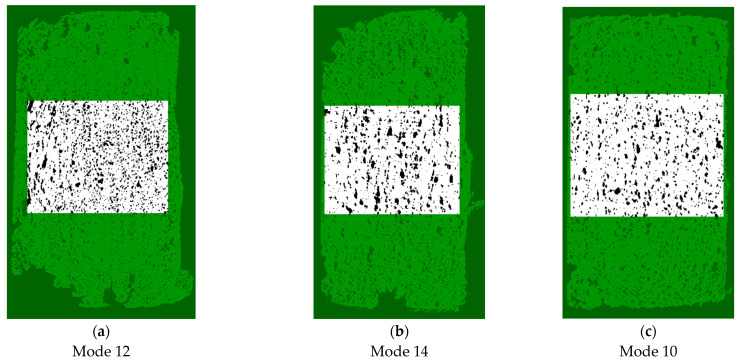
Visualizations of the areas (sections) selected for each of the modes to calculate the porosity levels for the images with sizes of 4.5–7.5 mm (**a**–**c**).

**Figure 17 polymers-16-02601-f017:**
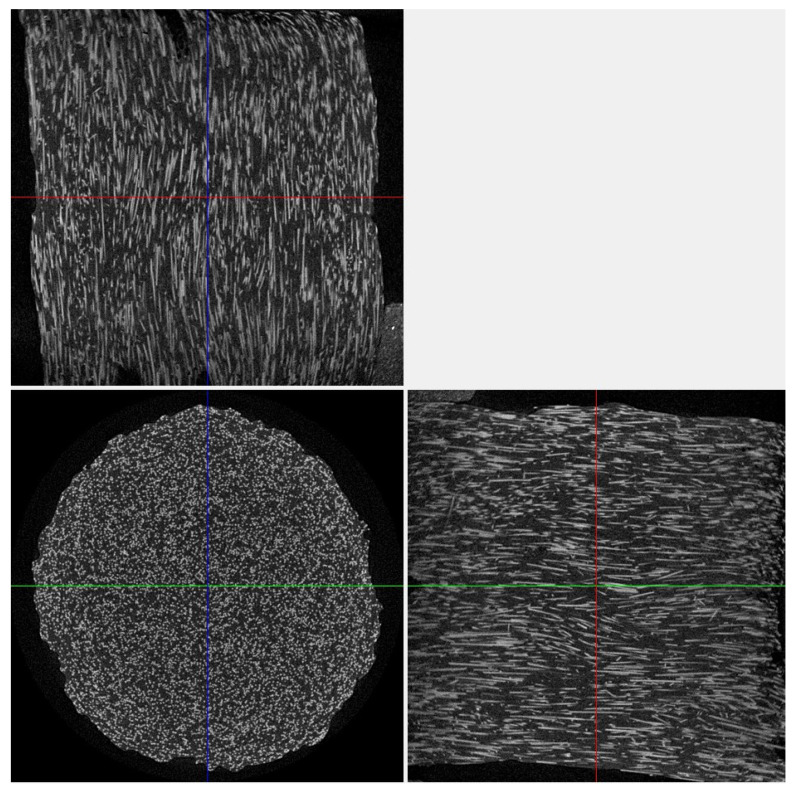
The orthogonal projections of an individual PEEK granule. Red denotes Z- axis section; Blue denotes X-axis section; Green denotes Y-axis section.

**Table 1 polymers-16-02601-t001:** The key characteristics of the ‘OREL-MT’ tomograph.

Characteristic	Value
X-ray machine	XWT 160-TC (X-RAY WorX, Garbsen, Germany)
Anode voltage, kV	20–160
Anode current, µA	5–1000
Focal spot, µm	1.4
Detector	PaxScan-2520V (Varian, Palo Alto, CA, USA)
Pixel size, µm	127
Detector operating area, mm	193 × 242
Detector size, pixels	1900 × 1516
Linear manipulator	MDrive Plus 42 (Intelligent Motion Systems, Inc., Marlborough, CT, USA)
Movement range, mm	0–400
Rotary system	MS208Ec (Physik Instrumente (PI) GmbH & Co., Karlsruhe, Germany)
Minimum angular step, degree	0.007

**Table 2 polymers-16-02601-t002:** The level numbers and the values of the Taguchi factors according to the L9 format.

Experiment Number	Levels and Values of (Technological) Factors		
	Extruder Temperature *T*_ext_, °C	Travel Speed *V*_3D_, mm/s	Extrusion Rate ν, Rpm
1	420 (1)	20 (1)	4.0 (1)
2	420 (1)	30 (2)	4.5 (2)
3	420 (1)	40 (3)	5.0 (3)
4	440 (2)	20 (1)	4.5 (2)
5	440 (2)	30 (2)	5.0 (3)
6	440 (2)	40 (3)	4.0 (1)
7	460 (3)	20 (1)	5.0 (3)
8	460 (3)	30 (2)	4.0 (1)
9	460 (3)	40 (3)	4.5 (2)

**Table 3 polymers-16-02601-t003:** The results of the tensile tests of the samples additively manufactured from the PEEK/30GF composite.

3D Printing Mode	Tensile Strength, MPa	Elastic Modulus, MPa	Elongation at Break, %
1	86.10 ± 6.12	2100 ± 216	4.8 ± 0.3
2	88.10 ± 6.65	2458 ± 346	4.7 ± 0.1
3	84.00 ± 2.72	2349 ± 232	4.8 ± 0.4
4	91.00 ± 6.29	2456 ± 232	4.7 ± 0.5
5	94.50 ± 4.39	2359 ± 162	5.1 ± 0.2
6	84.60 ± 2.19	2326 ± 268	4.6 ± 0.2
7	94.50 ± 1.53	2610 ± 173	4.8 ± 0.3
8	93.80 ± 2.75	2542 ± 8	5.2 ± 0.4
9	85.00 ± 1.44	2378 ± 357	4.3 ± 0.3

**Table 4 polymers-16-02601-t004:** Ranking of the input (control) parameters by their Δ influence degrees.

Property	Delta Parameter Δ (or L_max_ − L_min_, i.e., the Difference in the S/N Values between the Maximum and Minimum Factor Levels)		
	Extruder Temperature T_ext_	Extrusion Rate ν	Travel Speed V_3D_
Tensile strength	0.804	0.808	0.879
Elastic modulus	0.808	0.462	0.485
Elongation at break	0.175	0.561	0.853

**Table 5 polymers-16-02601-t005:** The 3D printing parameters of the laboratory experiments, in addition to those analyzed by the Taguchi method.

3D Printing Mode	Extruder Temperature T_ext_, °C	Travel Speed *V*_3D_, mm/s	Extrusion Rate ν, rpm
10	460	20	4
11	380	20	5
12	380	40	5
13	400	30	5
14	440	10	5
15	440	30	3.5

**Table 6 polymers-16-02601-t006:** The regression statistics of the experimental data.

Statistics	Predicted Property			Average
	Tensile Strength	Elastic Modulus	Elongation at Break	
Multiple R	0.573	0.243	0.445	0.4203
R-squared	0.329	0.059	0.198	0.1953
Normalized R-squared	0.299	0.018	0.163	0.1600
Standard error	8.846 MPa	378.091 MPa	0.594%	–
MSE	0.0983	0.1260	0.0914	0.1052
Multiple R	0.573	0.243	0.445	0.4203

**Table 7 polymers-16-02601-t007:** The minimum acceptable property values for the samples of the PEEK/30GF composite.

Property	Minimum Acceptable Value
Tensile strength, MPa	90
Elastic modulus, MPa	2400
Elongation at break, %	4.8

**Table 8 polymers-16-02601-t008:** The statistical characteristics of the selected models, the 3D printing parameters, and the corresponding predicted maximum property values for the samples of the PEEK/30GF composite.

ANN Type	Sample Size	MSE	R2	3D Printing Parameters			Mechanical Properties		
				Extruder Temperature T_ext_, °C	Travel Speed V_3D_, mm/s	Extrusion Rate ν, rpm	Tensile Strength, MPa	Elastic Modulus, MPa	Elongation at Break, %
RBFNN	66	0.0235	0.4543	460.40	25.07	4.10	98.85	2559.17	5.05
	66 + 54	0.0427	0.9837	454.54	30.91	4.59	100.51	2525.50	5.58
FFNN	66 + 132	0.0255	0.4124	500	18.64	3.85	99.55	2605.99	5.31
	66 + 54 + 240	0.0560	0.9788	453.36	13.92	4.51	95.02	2645.10	5.19

**Table 9 polymers-16-02601-t009:** The porosity levels for the AM samples, according to the micro-CT analysis.

Mode	VOI, mm^3^	Material Volume, mm^3^	Porosity, %
Mode 12	59.76	50.25	16
Mode 14	57.15	50.26	12
Mode 10	57.15	51.71	9.5

## Data Availability

The original contributions presented in the study are included in the article/Appendix A, further inquiries can be directed to the corresponding author.

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
