# Peer review of "Optimization of 3D Printing Parameters of High Viscosity PEEK/30GF Composites"

_polymers, 2024, doi:10.3390/polym16182601_

Round 1

Reviewer 1 Report

Comments and Suggestions for Authors

The topic of this manuscript is interesting. The experimental data cannot support the conclusion well. The authors should address the following comments before publishing in this journal.

1. The authors should state clearly why you focus on PEEK/30GF composite rather than 30 wt% carbon fiber. A summary of the advantages of this composite should be given in the introduction part.

2. The number of samples for the training of the artificial neural networks is not enough. Some more samples are needed to verify the model.

3. The processing parameters are not reasonable, because the mechanical properties of the samples have no big difference.

4. SEM images and FT-IR results have no meaning for this research because the authors use ANN to predict mechanical properties. 

Author Response

The topic of this manuscript is interesting. The experimental data cannot support the conclusion well. The authors should address the following comments before publishing in this journal.

Esteemed reviewer, we greatly appreciate your time, expertise, valuable comments and remarks while reading and analyzing the manuscript. We also thank you for the positive response of the conducted research. We have taken all your remarks into account when revising the manuscript. Please find below the line-by-line responses to all of your comments and remarks.

  1. The authors should state clearly why you focus on PEEK/30GF composite rather than 30 wt. % carbon fiber. A summary of the advantages of this composite should be given in the introduction part.

Esteemed reviewer, thank you for the relevant remark. In fact, we do have experience in studying PEEK based composites loaded with carbon fiber that ensures higher mechanical properties in the composites. However, the current study was mostly focused on technological properties of PEEK-based composites, i.e. 3D-printability. In this regard, the role of the filler material is not so critical in contrast to its weight fraction. Moreover, we aimed to employ lower cost commercially available granules of PEEK/GF30 in the view of potential industrial application of results. In doing so, the glass fibers were selected as the filler. We appreciate your idea on changing the filler for carbon fibers and shall do it in the forthcoming study.

  1. The number of samples for the training of the artificial neural networks is not enough. Some more samples are needed to verify the model.

Esteemed reviewer. In fact, you are absolutely correct. In order to train the ANN, the large sample size is required when the classification or recognition problem is to be solved. This is of actuality as well, when enough experimental data are available. However, when the approximation problem or regression model reconstruction are to be solved, the increase of sample size is not always accompanied by increasing the simulation accuracy. This was shown in our previous paper on the topic.

In addition, for variation of several technological (input) parameters the design of experiment is to be carried out in order to avoid conducting full factor study. This was done through application of Taguchi method. However, this was not surely enough. The idea of expanding the sample size was described in the paper. This made it possible to find out the optimum area in the parameters’ space. In addition, in our previous study on ultrasonic welding, it has been shown that at using similar types of ANN the satisfactory sample size might be at least 30-50. We have exceeded the size in this paper.

We must confess that verification of constructed models is to be conducted as well. However, the area of optimum parameters was well-supported by the already conducted experimental data.

  1. The processing parameters are not reasonable, because the mechanical properties of the samples have no big difference.

Esteemed reviewer, thank you for the relevant remark. You are correct, the optimization of studied technological parameters did not allow to drastically increase the mechanical properties. However, we did vary those 3D-printing parameters that are discussed in the literature and whose changing can be done directly in our pilot 3D printing machine. This made it possible to compare our results with the literature. The reason was related to high viscosity of PEEK/30GF blend. We could not avoid high porosity of the 3D printed composites by varying the input parameters. According to the authors, the best way is application of post build processing. We do focus on such experiments these days. In addition, the ANN simulation makes it possible both to find out predicted local values of optimum parameters, and to construct their range estimates.

  1. SEM images and FT-IR results have no meaning for this research because the authors use ANN to predict mechanical properties. 

Esteemed reviewer, thank you for the relevant comment. The SEM and FTIR data are given in order to characterize the structure of 3D printed composites PEEK/30GF. This made it possible to reveal the role of porosity and prove the maintenance of chemical structure at increased 3D printing temperatures. We ask you not to insist on this remark, since the information provided is of importance in the materials science oriented journal.

Esteemed reviewer, let us thank you once again for the relevant comments and remarks. They did help us to improve the manuscript and make it more understandable. 

Reviewer 2 Report

Comments and Suggestions for Authors

The study optimized the input parameters for 3D printing with PEEK-based composites using the Taguchi method and finite element methods. The results demonstrated improved mechanical properties. Additionally, an Artificial Neural Network (ANN) was trained, showing that the parameters closely aligned with those identified by the Taguchi method. The article is well-organized and well-written. However, some minor suggestions are provided to better meet publication requirements.

1. On page 1 line 44, Providing additional details will help readers from various fields gain a deeper understanding of the background of the work. For the 30 wt% mentioned, which represents the total of CCFs and GFs, could you specify the composition of each component individually?

2. On page 2, line 52, what are the other 90 wt.% or any reference for it?

3. On page 3, line 63, Please list other methods here and highlight the reason why the Taguchi is widely applied. Such as ‘compared to the other mehtods […], the Taguchi method is widely used due to … [14]’

4. On page 3, line 72, please exemply how the train data is collected in one sentece, rather than listing a reference.  

5. On page 5, Table 2, it is confusing to list in format 1/420. how about change the format into ‘420, 1’ 

6. On page 6, Figure 1, the x labels should be written on the bottom axis. Place the title for each plot.

7. On page 8, Figure 2, 9 sets are listed in Table 2. A statement how the plots correlate to Table 2 is needed. More detailed statement should be given in the caption.

8. On page 13, Figure 6, it would be better to show the minimal values on each axises.

Author Response

The study optimized the input parameters for 3D printing with PEEK-based composites using the Taguchi method and finite element methods. The results demonstrated improved mechanical properties. Additionally, an Artificial Neural Network (ANN) was trained, showing that the parameters closely aligned with those identified by the Taguchi method. The article is well-organized and well-written. However, some minor suggestions are provided to better meet publication requirements.

Esteemed reviewer, we greatly appreciate your time, expertise and valuable comments and remarks while reading and analyzing the manuscript. We also thank you for the positive response of the conducted research. We have taken all your remarks into account when revising the manuscript. Please find below the line-by-line responses to all of your comments and remarks.

  1. On page 1 line 44, Providing additional details will help readers from various fields gain a deeper understanding of the background of the work. For the 30 wt% mentioned, which represents the total of CCFs and GFs, could you specify the composition of each component individually?

- Esteemed reviewer, thank you for the relevant comment. We have been analyzing the studies on various PEEK-based compositions for a long time and the conclusion on typical filler weight fraction have been made on the basis of various observations. If you don’t mind, we would cite a couple of relevant references that support the idea of loading PEEK with 30% of CF and GF.

  1. On page 2, line 52, what are the other 90 wt.% or any reference for it?

- Esteemed reviewer, the rest of the filament is PEEK. We would rather not give additional references, since it might be occasionally treated as an advertisement. The text was modified a little bit.

  1. On page 3, line 63, Please list other methods here and highlight the reason why the Taguchi is widely applied. Such as ‘compared to the other mehtods […], the Taguchi method is widely used due to … [14]’

- Esteemed reviewer, thank you for the relevant remark. It has been taken into account while revising the manuscript. The due corrections are highlighted with cyan marker.

  1. On page 3, line 72, please exemply how the train data is collected in one sentece, rather than listing a reference.  

- Esteemed reviewer, we are grateful for the relevant remark. The required explanation has been added to the text of the revised manuscript.

  1. On page 5, Table 2, it is confusing to list in format 1/420. how about change the format into ‘420, 1’ 

- Esteemed reviewer, thank you for the brilliant idea! The recommended correction has been made.

  1. On page 6, Figure 1, the x labels should be written on the bottom axis. Place the title for each plot.

- Esteemed reviewer. We appreciate the valuable piece of advice. Since figure 1 illustrates a tripled diagram rather than a graph, it would be better not to put it on the horizontal axis, otherwise it would be somehow confusing. The Taguchi diagrams of this kind are conventionally presented in this way.

  1. On page 8, Figure 2, 9 sets are listed in Table 2. A statement how the plots correlate to Table 2 is needed. More detailed statement should be given in the caption.

- Esteemed reviewer, thank you for the relevant remark. The due explanation has been added to the revised manuscript.

  1. On page 13, Figure 6, it would be better to show the minimal values on each axises.

- Esteemed reviewer, we are grateful for the relevant remark. The due correction has been made in figure 6.

Esteemed reviewer, let us thank you once again for the relevant comments and remarks. They did help us to improve the manuscript and make it more understandable.

Round 2

Reviewer 1 Report

Comments and Suggestions for Authors

The authors did not offer valuable revisions in the revised manuscript. I recommend to reject it completely.

Author Response

- Esteemed reviewer, we are really sorry for making an impression of tiny modification of the revised manuscript. Instead of embedding the corrections in the revised text, we focused on detailed responses just in the reviewer’s form. Anyway, we greatly appreciate your time, expertise, valuable comments and remarks while reading and analyzing the manuscript. We did take into account all your remarks when repeatedly revised the manuscript. Please find below the line-by-line responses to all of your comments and remarks. The correction made in the revised manuscript following your remarks have been highlighted with green marker. They are given below as well.

  1. The authors should state clearly why you focus on PEEK/30GF composite rather than 30 wt. % carbon fiber. A summary of the advantages of this composite should be given in the introduction part.
  • Esteemed reviewer, thank you for the relevant remark. The following text has been added to the Introduction section of the revised manuscript.

The selection of glass fibers as a filler material was motivated by their: i) availability of commercial grade feedstocks (granules); ii) lower price; iii) easiness of processing; iv) large interphase adhesion, and v) possibility of substantially improving mechanical properties of PEEK. Their input in increasing mechanical properties of PEEK-based composites is a little bit lower in contrast with carbon fibers; however, in terms of affecting 3D-printability they are quite comparable.

  1. The number of samples for the training of the artificial neural networks is not enough. Some more samples are needed to verify the model.
  • Thank you for the comment. The text of the revised manuscript was extended in order to verify the model. The additional figure 8 was added. The following explanation was added as well.

In order to estimate the influence of training sample size, its effect on accuracy of model construction for the FFNN was carried out at variation of the number of vectors of input parameters as well as amount of synthesized data, Figure 8. During synthesis of the models, the mean square error (MSE) was utilized as the optimization function. It allowed to characterize the deviation of the model from the training set. The mean absolute deviation (MAD) of the normalized model values over a priory values not taken/specified/determined in the training set was utilized for characterizing the prognosis capability of the models. The verification of the FFNN models over the data of a priory knowledge planes has shown quite different dependence: the presence of MAD maximum at the sample size of 69 modes (vectors of the input parameters) and 120 vectors of the corresponding mechanical properties. At the same time, for the FFNN model, the training sample was enlarged by synthesized data with the characteristics similar to those at the first stage (see above). The latter has increased the prognosis capabilities of the developed models.

  1. The processing parameters are not reasonable, because the mechanical properties of the samples have no big difference.

Thank you for the relevant remark. You are absolutely correct; there is no big difference. However, the values of mechanical properties are affected by 3D printing parameters. This means that: i) they were close to optimal values in the studied range; ii) high viscosity of the feedstock does not make it possible to substantially reduce the porosity (being revealed by SEM micrographs and numerically characterized by the CT). We did vary the 3D printing parameters, being characteristic for the most of the papers on FDM. The result is as it is. We do consider it as an important outcome of the study. Some discussion on this issue has been added to the revised manuscript (the following paragraph has been added to the Discussion Section).

It should be noticed, that the optimization of studied technological parameters did not allow to drastically increase the mechanical properties, including with the use of results of the ANN simulation. However, it does not mean that they did not affect the structure and functional properties. The above cited literature on FDM of PEEK and PEEK based composites did focus on variation of the same 3D-printing parameters [29]. However, use of low viscosity polymers allows reducing the porosity and attain the wider range of variation of mechanical properties. Thus, the within the current paper the authors were able to compare the result of this study with those reported in the relevant papers [11, 12, 29]. Since it was not possible to avoid high porosity of the 3D printed PEE/30GF composites by varying the input parameters with the use of the available 3D-printer, the prospect might be related to application of post build processing, i.e. ultrasonic compaction with the use of US-welding machine. In these regard, the developed approach to ANN simulation will be of efficiency both to find out predicted local values of optimum parameters, and to construct estimates of their range.

  1. SEM images and FT-IR results have no meaning for this research because the authors use ANN to predict mechanical properties.

Esteemed reviewer, thank you for the relevant comment. Half of the SEM images were moved to the Supplementary, as well as all the FTIR data. In addition, the title of the manuscript has been modified according to the academic editor recommendation. In the corrected form, the SEM data on microstructure has become of actuality. At least, the FEM part is based on calculation of effect of porosity, while it was evident only at SEM micrographs.

- The text of conclusions has been corrected in order to fit well the main outcomes of the study.

The following correction has been made.

The effect of the 3D printing parameters on the structure and the mechanical properties of the samples manufactured by the FDM method from the high-viscosity PEEK/30GF composite feedstock was investigated. It was shown that the formed macro- and microstructures limited the achievement of their high levels. In particular, the following conclusions were drawn.

  1. The high viscosity of the molten polymer contributed to the great porosity levels of the AM samples that was proven in details by the SEM micrographs. The presence of pores reduced the elastic moduli below 3 GPa, while the elongation at break values of ~5% were significantly higher than those for similar compression molded (hot-pressed) composites.
  2. By computer simulation using two types of ANNs, the optimal combinations of the 3D printing parameters were determined for the PEEK/30GF composite: Extruder temperature: 450¸460 ᵒС, Extrusion rate: 4.1¸59 rev/min, travel speed; 13.92¸30.91 mm/s. When an ultra-small sample was utilized, the application of a priory knowledge made it possible to achieve the appropriate simulation accuracy (MSE~ 0.06). In doing so, the RBFNN were able to construct more realistic models, while for the FFNN the reasonable limitation of it its size was implemented.
  3. An interpretation of the differences in the results predicted using the RBFNN and FFNN models was proposed based on their operation principles. Generally, the determined optimal values were consistent with those obtained by the Taguchi method, and physically corresponded to the assumption formulated as the null hypothesis: higher quality of AM products from the PEEK/30GF composite is ensured by maximizing material feeding into the 3D-printing zone by: reducing the V3D travel speed of the moving extruder head with increasing both the Text extruder temperature and the n extrusion rate.
  4. The effect of porosity on the mechanical properties of the additively manufactured PEEK/30GF composites were assessed by implementing the FEM-based models of small, large and mixed pores. The obtained results made it possible to explain the experimentally revealed relatively low level of strength.

Esteemed reviewer, let us thank you once again for the relevant comments and remarks. They did help us to improve the manuscript and make it more understandable.

Round 3

Reviewer 1 Report

Comments and Suggestions for Authors

It can be accepted after this revision.